# Adherens junctions limit septate junction length in *Drosophila* midgut enterocytes but are not required for polarity

Cátia A. Carvalho, Mihoko A. Tame and Daniel St Johnston\*

## ABSTRACT

Adherens junctions formed by E-cadherin adhesion complexes play central roles in the organisation and apical-basal polarisation of both mammalian and insect epithelia. Here, we investigate the function of the components of the E-cadherin adhesion complex in the *Drosophila* midgut epithelium, which establishes polarity by a different mechanism from other fly epithelia and has an inverted junctional arrangement in which the adherens junctions lie below the septate junctions. Unlike other epithelial tissues, loss of E-cadherin, Armadillo (β-catenin) or α-catenin has no effect on the polarity or organisation of the adult midgut epithelium. This is not due to redundancy with other cadherins, as enterocytes lacking E-cadherin, N-cadherin and CadN2 still polarise normally. However, E-cadherin (*shg*) and *armadillo* mutants have expanded septate junction domains and shorter lateral domains below the septate junctions, indicating that E-cadherin adhesion complexes limit the basal extent of the septate junctions. Thus, Cadherin-mediated adhesion is dispensable for apical-basal polarity and epithelial organisation in the *Drosophila* midgut, in contrast to all other epithelia that have been studied so far, but it is required to define the size of the septate junctions and cell height.

KEY WORDS: E-cadherin, N-cadherin, Apical-basal polarity, Armadillo, Triple-knockout clones, Cell height

## INTRODUCTION

A defining feature of epithelial cells is their ability to adhere to each other laterally to form epithelial sheets and tubes. In most epithelia, this lateral adhesion depends on adherens junctions formed by the E-cadherin adhesion complex, composed of the homophilic adhesion protein, E-cadherin (encoded by the *shotgun* gene in *Drosophila*) and the cytoplasmic adaptor proteins, β-catenin (Armadillo in *Drosophila*), α-catenin and p120-catenin (p120ctn) (Harris and Tepass, 2010; van Roy and Berx, 2008; Tepass et al., 2001). E-cadherin binds in trans through opposing extracellular cadherin repeats (EC1–EC4) in neighbouring cells and also interacts with other E-cadherin molecules in cis to form clusters (Mège and Ishiyama, 2017). p120ctn binds to the cytoplasmic (juxtamembrane) domain of E-cadherin to increase adhesion and

The Gurdon Institute and Department of Genetics, University of Cambridge, Tennis Court Rd, Cambridge CB2 1QN, UK.

\*Author for correspondence (ds139@cam.ac.uk)

C.A.C., 0000-0001-7810-9127; M.A.T., 0000-0001-9214-0390; D.S., 0000-0001-5582-3301

antagonise E-cadherin endocytosis, and this is essential for robust cell–cell adhesion in vertebrates but is dispensable in *Drosophila* (Davis et al., 2003; Ireton et al., 2002; Pacquelet and Rørth, 2005; Pacquelet et al., 2003; Thoreson et al., 2000). The E-cadherin cytoplasmic domain contains a more-C-terminal binding site for β-catenin, which in turn recruits α-catenin, which links the complex to the F-actin cytoskeleton (Desai et al., 2013; Harris and Tepass, 2010).

The establishment of E-cadherin-dependent adherens junctions is intimately linked to the development of polarised epithelial cells in vertebrates. For example, expressing E-cadherin in unpolarised fibroblasts is sufficient to induce the polarised trafficking of basolateral proteins to cell contact regions (McNeill et al., 1990). Furthermore, loss of E-cadherin in mouse embryos leads to a failure to form the trophectoderm epithelium and arrests development, whereas the specific deletion of E-cadherin in the skin not only removes the adherens junctions but causes defects in the overlying tight junctions and a loss of barrier function (Larue et al., 1994; van Roy and Berx, 2008; Tunggal et al., 2005). Similarly, siRNA-mediated knockdown of α-catenin in MDCK cells disrupts polarity completely (Capaldo and Macara, 2006).

The organisation of junctions in most invertebrate epithelia is inverted compared to that in vertebrates, as the adherens junctions concentrate in zonula adherens at the top of the lateral domain above more-basal septate junctions, which are analogous to vertebrate tight junctions and perform the barrier function (St Johnston and Ahringer, 2010; Tepass et al., 2001). Nevertheless, loss of E-cadherin complex components has a similar effect on epithelial organisation and polarity to that in vertebrates. It is not possible to remove all E-cadherin from the *Drosophila* embryo because there is a large maternal contribution that is required for normal oogenesis (Tepass et al., 2001). However, in embryos derived from germline clones of strong β-catenin mutants (*arm*) that also lack zygotic β-catenin, embryonic tissues lose their epithelial characteristics at gastrulation, producing unpolarised mesenchymal cells (Cox et al., 1996). A similar phenotype is observed with RNAi-mediated knockdown of both maternal and zygotic α-catenin (Sheppard et al., 2022). The role of E-cadherin adhesion complexes has also been examined in the follicular epithelium that surrounds developing female germline cysts (Tepass et al., 2001). E-cadherin mutant clones cause various developmental defects, but do not disrupt epithelial organisation or polarity because of redundancy with N-cadherin (Godt and Tepass, 1998; González-Reyes and St Johnston, 1998; Pacquelet and Rørth, 2005; Tanentzapf et al., 2000). By contrast, removing all Cadherin-based adhesion by generating *armadillo*- and α-catenin-null mutant clones leads to a loss of epithelial organisation and disrupted apical-basal polarity (Bonello et al., 2021; Sarpal et al., 2012; Tanentzapf et al., 2000).

Cadherin adhesion complexes contribute to epithelial polarity through several mechanisms. First, by linking sites of cell adhesion to the actin cytoskeleton, they generate robust intercellular junctions that can resist the forces exerted during morphogenesis (Harris and

Tepass, 2010; Lecuit and Yap, 2015). Second, the key polarity factor Bazooka (Par-3) is recruited to adherens junctions through direct interactions with E-cadherin and β-catenin, where it is thought to form phase-separated condensates that provide a barrier between the apical and lateral domains (Bonello et al., 2021; Buckley and St Johnston, 2022; Kono et al., 2019; Liu et al., 2020; Wei et al., 2005). Third, RhoGAP19D is recruited to adherens junctions along the lateral domain, where it inactivates Cdc42, thereby restricting the activity of the apical Cdc42–Par-6–aPKC complex to the apical side of the cell (Fic et al., 2020).

The intestinal epithelium of the *Drosophila* midgut has a different organisation from that in other fly epithelia as it forms septate junctions at the top of the lateral domain above adherens junctions, a junctional arrangement that resembles that in vertebrates (Baumann, 2001). Furthermore, the midgut enterocytes do not express key polarity factors, such as Bazooka (Par-3), and their apical-basal polarisation does not require any of the canonical epithelial polarity factors that polarise other *Drosophila* epithelia (Chen et al., 2018). These differences might reflect the endodermal origin of the midgut, or the fact that, unlike other fly epithelia, enterocytes polarise in a basal to apical direction as they differentiate from basally located enteroblasts (EBs) and integrate into the epithelium (Chen and St Johnston, 2022; Galenza et al., 2023). The distinct organisation of the midgut epithelium raises the question of what roles E-cadherin adhesion complexes play in enterocyte polarisation. E-cadherin has previously been shown to mediate adhesion between the daughters of basal intestinal stem cell (ISC) divisions, where it acts to delay cell detachment to allow Delta–Notch signalling to specify one daughter cell as an EB (Maeda et al., 2008). In addition, downregulation of E-cadherin in apoptotic enterocytes triggers the release of epidermal growth factors to stimulate compensatory divisions of the ISCs,

thereby maintaining homeostasis (Liang et al., 2017). E-cadherin must also be removed from the apical surface of integrating EBs to establish an apical membrane initiation site (AMIS) (Chen and St Johnston, 2022). However, the effects of E-cadherin complex depletion on enterocyte polarity and differentiation have not been analysed. Here, we set out to investigate the role of E-cadherin complex components in this process and demonstrate that adherens junctions are dispensable for enterocyte polarity and epithelial integrity but play a role in defining the size of the septate junctions and the extent of the lateral domain.

## RESULTS
### Adherens junction components localise below the septate junctions

To verify the composition of adherens junctions in the *Drosophila* midgut, we used antibodies and protein trap lines to examine the localisation of the E-cadherin adhesion complex components (Fig. 1A). Endogenously tagged E-cadherin–GFP was highly expressed in ISCs and EBs, as previously reported (Chen and St Johnston, 2022; Huang et al., 2009; Maeda et al., 2008). E-cadherin was also expressed in enterocytes, albeit at lower levels, and localised strongly along the region of the lateral membrane below the septate junctions, which were marked by Coracle (Fig. 1B). In cross-sections through the epithelium, E-cadherin was distributed in diffraction-limited puncta along the cell membranes, which presumably correspond to spot adherens junctions. E-cadherin was not restricted to the lower portion of the lateral domain, however, as low levels were also found throughout the septate junction region (Fig. 1C). Antibody staining for β-catenin (Armadillo) and an α-catenin protein trap line displayed similar distributions to E-cadherin (Fig. 1D,E). By contrast, p120ctn was not expressed at

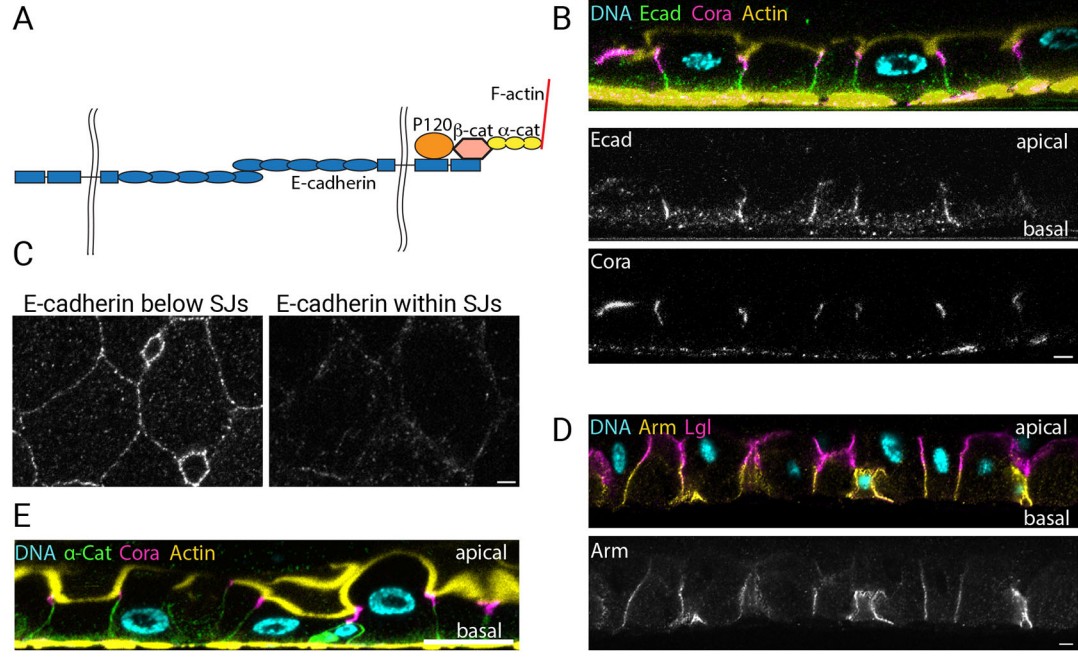

**Fig. 1. E-cadherin, β-catenin and α-catenin colocalise beneath the septate junctions in the *Drosophila* midgut.** (A) A diagram showing the components of E-cadherin adhesion complexes. The black wavy lines indicate the opposing plasma membranes. (B) The localisation of endogenously tagged E-cadherin-GFP (green) along the apical-basal axis of the midgut epithelium, stained for DNA (DAPI; blue), Coracle (magenta) and F-actin (yellow). (C) A horizontal view of E-cadherin at the level of the adherens junctions (left) and at the level of the septate junctions (SJs; right). (D) β-catenin localisation (yellow) along the apical-basal axis of the midgut epithelium, stained for DNA (DAPI; blue) and Lgl (magenta). (E) An apical-basal confocal section showing the localisation of α-catenin protein trap line (green) stained for DNA (DAPI; blue), Coracle (magenta) and actin (yellow). Images in this figure are representative of two repeats. Scale bars: 20 µm.

detectable levels in the midgut, given that a monoclonal antibody that recognises p120ctn in other tissues gave only non-specific staining of the septate junctions, which was still present in *p120ctn* homozygous clones (Iyer et al., 2019; Magie et al., 2002). These results confirm that adherens junctions form below the septate junctions in the midgut, which is the opposite way round to what is seen in other *Drosophila* epithelia but resembles the junctional arrangement in vertebrates (Chen et al., 2018).

## The localisations of adherens junction components are mutually dependent

The core components of the adherens junctions rely on mutually dependent interactions to localise at the plasma membrane. Although intercellular adhesion is nucleated by the trans-homophilic interaction between E-cadherin extracellular domains, the efficient trafficking of E-cadherin from the endoplasmic reticulum to the plasma membrane depends on the binding of its cytoplasmic tail to β-catenin (Chen et al., 1999; Curtis et al., 2008; Tanentzapf et al., 2000). Conversely, loss of E-cadherin impairs the targeting of β-catenin and α-catenin to the plasma membrane, and α-catenin depletion decreases both E-cadherin and β-catenin localisation at cell–cell contact sites (Bonello et al., 2021; Capaldo and Macara, 2006; Pacquelet and Rørth, 2005; Sarpal et al., 2012). To investigate whether these mutual dependencies are maintained in the *Drosophila* midgut, we generated homozygous clones for null alleles of β-catenin ($arm^{YD35}$) E-cadherin ($shg^{R69}$) and α-catenin ($\alpha\text{-}cat^1$) using the 'mosaic analysis with a repressible cell marker' (MARCM) system (Godt and Tepass, 1998; Lee and Luo, 2001; Peifer and Wleschaus, 1990; Sarpal et al., 2012). As expected, $arm^{YD35}$ homozygous enterocytes lacked β-catenin staining at the membrane, which was outlined by staining for the septate junction protein Mesh (Fig. 2A). A plot of the β-catenin signal intensity across the lateral membrane in the $arm^{YD35}$/+ heterozygous control showed a clear peak at the cell junction, whereas the homozygous mutant clone showed a uniform level of staining (Fig. 2B). Given that $arm^{YD35}$ is a null allele that produces no protein (Peifer and Wleschaus, 1990), this residual signal might correspond in part to perdurance of the wild-type protein after clone induction, but is most likely background staining (Fig. 2B,G). GFP-labelled $shg^{R69}$ homozygous enterocytes also lost β-catenin staining from the plasma membrane (Fig. 2C). β-catenin intensity plots showed no peak at the adherens junction, indicating that E-cadherin is required for all β-catenin localisation at the membrane (Fig. 2D,G). Furthermore, the cytoplasmic signal remained at similar levels to that in $shg^{R69}$/+ heterozygous cells, suggesting that the β-catenin that fails to localise to junctions is degraded, most probably by the APC–Axin destruction complex (Fig. 2H) (Clevers, 2006). Similar results were observed with additional *shotgun* and *armadillo* alleles (Fig. S1A–F). Finally, mutant enterocytes lacking α-catenin displayed reduced levels of β-catenin at the lateral membrane, but still had a clear enrichment at the junctions (Fig. 2E,G). β-catenin intensity plots across mutant enterocytes revealed that there was a more uniform distribution of β-catenin throughout the cell, with slightly higher cytoplasmic levels than in the heterozygous controls (Fig. 2F,H). Thus, the mutual dependencies between the components of the E-cadherin adhesion complex are similar to those in other tissues, despite the altered junctional arrangement in the midgut.

## Adherens junctions are not required for the establishment of apical-basal polarity

Unlike other *Drosophila* epithelia, the apical-basal axis of *Drosophila* midgut is established in a basal-to-apical direction as EBs integrate into the epithelium, and the adherens junctions

therefore form first before the more apical septate junctions (Chen and St Johnston, 2022; Chen et al., 2018). Given the key role of adherens junctions in the apical-basal polarity of other epithelia in both vertebrates and *Drosophila*, we investigated whether loss of the E-cadherin adhesion complex would disrupt EB or enterocyte integration or polarisation. We generated mutant stem cell clones lacking each adherens junction component and examined their phenotypes 3–15 days after clone induction to reduce the perdurance of pre-existing wild-type protein and to ensure that the cells lacked the protein of interest from birth. Loss of E-cadherin did not prevent mutant EBs from integrating into the epithelium and differentiating into enterocytes (Fig. 3A,A'). Longitudinal views showed that $shg^{R69}$/$shg^{R69}$ enterocytes had well-defined apical domains contacting the gut lumen, localised Canoe normally at the apical-lateral boundary and were properly attached to the basement membrane (Fig. 3A''). Like $shg^{R69}$/$shg^{R69}$ cells, $arm^{YD35}$/$arm^{YD35}$ mutant cells integrated into the epithelium and formed a normal apical brush border, marked by the localisation of atypical protein kinase C (aPKC) and septate junctions labelled by Coracle (Fig. 3B–B''). α-catenin-null cells also integrated and established a wild-type apical-basal axis, with a fully-formed basal labyrinth, marked by Nervana (Nrv) (Fig. 3C–C''). To verify these results, we knocked down the core components of the E-cadherin complex by RNAi in enterocytes throughout the entire midgut. Downregulation of E-cadherin using two different RNAi lines led to a marked reduction in β-catenin localisation to cell junctions but did not cause any apical-basal polarity defects, as previously reported (Liang et al., 2017) (Fig. S2A–C). Downregulation of β-catenin or α-catenin also had no effect on tissue organisation or polarity (Fig. S2A,D,E). Altogether, these results demonstrate that the core components of the adherens junctions are not required for either the establishment or maintenance of apical-basal polarity in the midgut epithelium, unlike all other epithelia that have been examined so far.

## E-cadherin and β-catenin regulate enterocyte shape independently of α-catenin

Although the adherens junctions are not required for enterocyte polarity, they could still play a role in regulating cell shape and cytoskeletal organisation, as they do in other epithelia (Bonello et al., 2021). To explore how E-cadherin-based adhesions modulate enterocyte morphology, we measured the circularity of mutant cells at the basal side as a proxy for the degree to which they adopted the normal polygonal shape of midgut epithelial cells. Enterocytes mutant for E-cadherin (0.77±0.11, mean±s.d.) became significantly more rounded than their heterozygous neighbours (0.70±0.12) (Fig. S3A,D). Similarly, loss of β-catenin (0.77±0.14) also significantly reduced the polygonal shape of the enterocytes compared to that for the controls (0.70±0.11) (Fig. S3B,D). Enterocytes mutant for α-catenin (0.76±0.13) were also rounder than the neighbouring enterocytes (0.71±0.12), but this difference was not statistically significant (Fig. S3C,D). Altogether, these data indicate that the loss or reduction in adhesion in the basal portion of the lateral domain causes mutant cells to become less polygonal.

## E-cadherin and β-catenin limit the basal extent of the septate junctions

We next examined the contribution of the adherens junctions to lateral domain identity. The lateral domain of the enterocytes comprises two sub-domains – an apical septate junction domain and a more-basal domain that normally contains adherens junctions, which we will refer to as the sub-septate junction domain. The septate junction domain was significantly extended in enterocytes

Journal of Cell Science

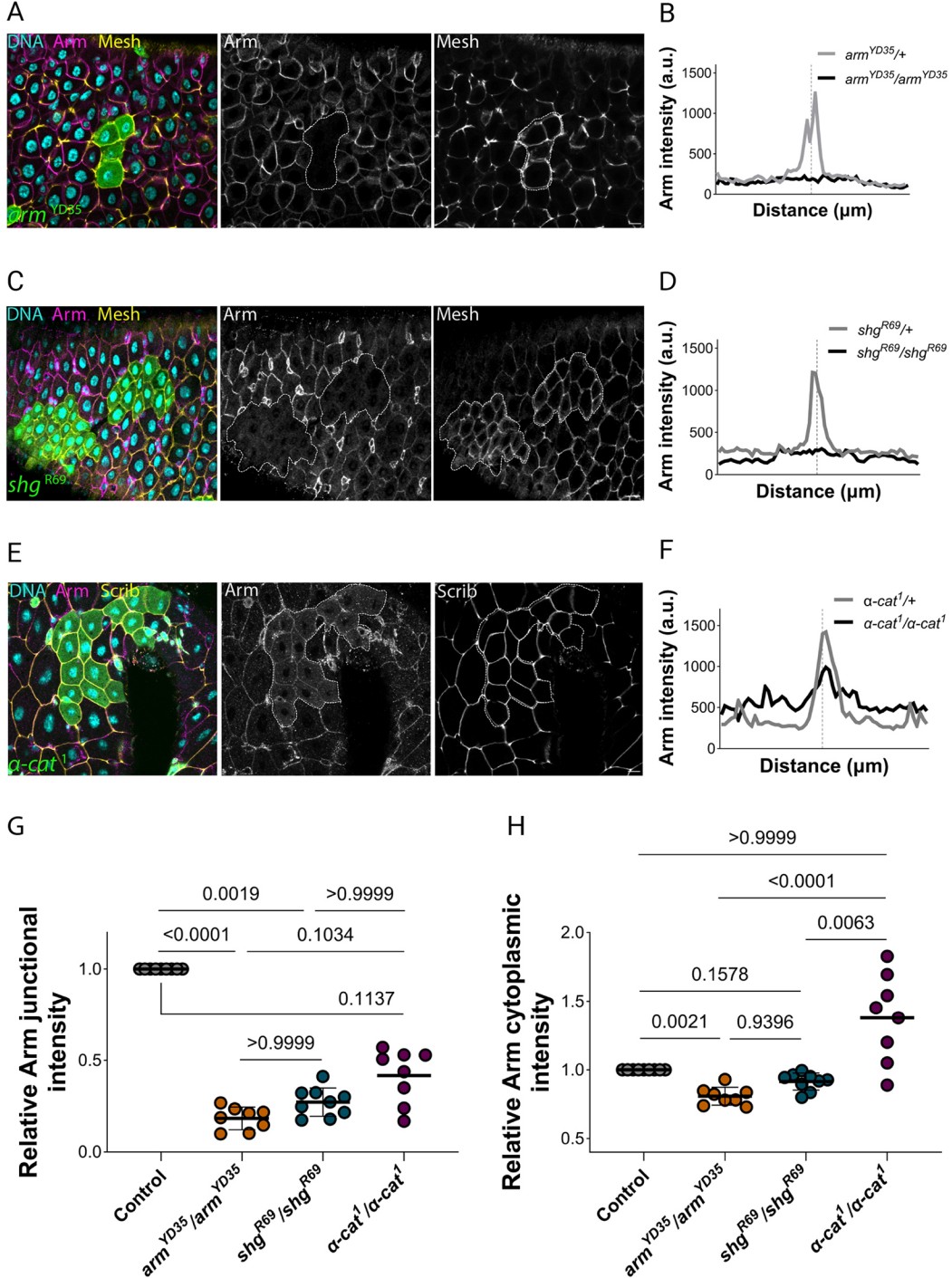

**Fig. 2. The adherens junctions core components are mutually dependent on their membrane localisation.** (A) A horizontal section through the basal region of a midgut containing a clone of $arm^{YD35}$ homozygous cells (GFP-positive; green), stained for DNA (DAPI; blue), Armadillo (magenta) and Mesh (yellow). (B) Plots of Armadillo signal intensity across two cells, centred at the position of the cell–cell boundary (dotted vertical line). The black line shows data from $arm^{YD35}$ homozygous enterocytes and the grey line, data from $arm^{YD35}/+$ heterozygous enterocytes. (C) A horizontal section through the basal region of the midgut containing a $shg^{R69}$ homozygous clone marked by GFP expression (green), stained for DNA (DAPI; blue), Armadillo (magenta) and Mesh (yellow). Clone boundaries in A–C are outlined with a white dashed line. (D) Armadillo intensity plots in $shg^{R69}$ homozygous enterocytes (black line) and $shg^{R69}/+$ heterozygous enterocytes (grey line). (E) A horizontal section through the basal region of the midgut containing an $\alpha$-$cat^1$ mutant clone (GFP) stained for DNA (DAPI; blue), Armadillo (magenta) and Scribble (yellow). (F) Armadillo intensity plots in $\alpha$-$cat^1$ homozygous (black line) and heterozygous (grey line) cells. Data in A–F are representative of five repeats for the $arm^{YD35}$ and $shg^{R69}$ experiments, and four repeats for the $\alpha$-$cat^1$ experiments. a.u., arbitrary units. (G) Measurements of Armadillo junctional intensity at the adherens junctions in heterozygous control cells (standardised mean=1, 9 midguts, 115 cell–cell junctions), $arm^{YD35}/arm^{YD35}$ (mean=0.18±0.06, 8 midguts, 91 cell–cell junctions), $shg^{R69}/shg^{R69}$ cells (mean=0.27±0.08, 9 midguts, 140 cell–cell junctions) and $\alpha$-$cat^1/\alpha$-$cat^1$ cells (mean=0.42±0.15, 8 midguts, 103 cell–cell junctions). Each circle in the plot shows the mean intensity for a single gut. (H) A superplot showing Armadillo intensity in the cytoplasm relative to heterozygous control cells in the same sample (standardised mean=1, 9 midguts, 93 cells), $arm^{YD35}/arm^{YD35}$ (mean=0.81±0.07, 8 midguts, 70 cells), $shg^{R69}/shg^{R69}$ (mean=0.91±0.06, 9 midguts, 119 cells) and $\alpha$-$cat^1/\alpha$-$cat^1$ (mean=1.38±0.32, 8 midguts, 92 cells) enterocytes. Each circle in the plot shows the mean intensity for a single gut. Error bars in G,H are mean±s.d. $P$-values shown calculated with a Shapiro–Wilk normality test followed by a Kruskal–Wallis with Dunn's post hoc test. Scale bars: 10 µm.

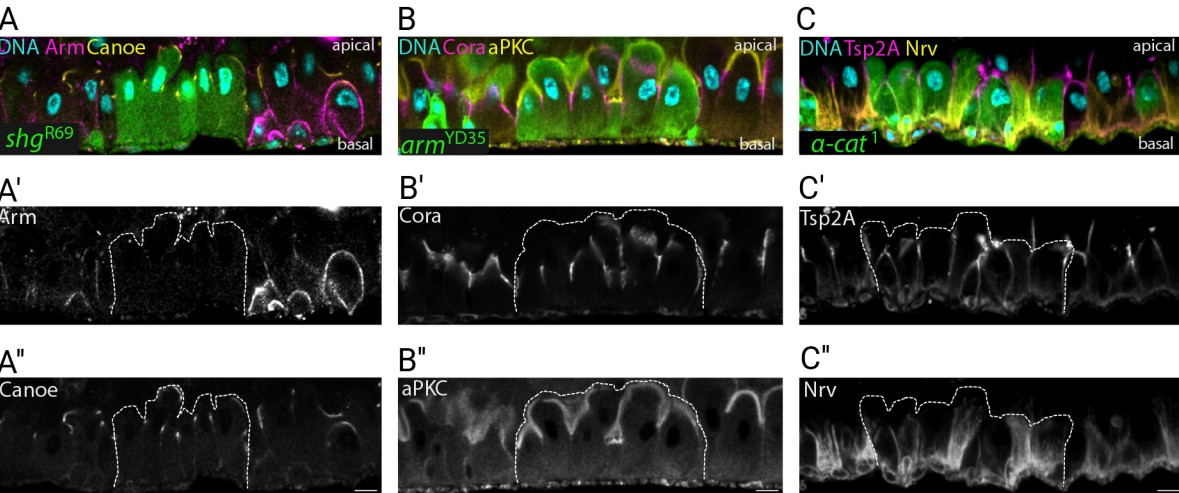

**Fig. 3. Adherens junctions are not required for apical-basal polarity in the midgut.** (A) An apical-basal section showing a $shg^{R69}$ homozygous clone (GFP; green) stained for DNA (DAPI; blue) (A), Armadillo (magenta and A′) and Canoe (yellow and A″). (B) An apical-basal section showing an $arm^{YD35}$ homozygous clone (GFP; green), stained for Coracle (magenta and B′) and aPKC (yellow and B″). (C) An $\alpha\text{-}cat^1$ homozygous clone (GFP; green) stained for Tsp2A (magenta and C′) and Nervana (yellow and C″). White dashed lines outline the clones in A′, A″, B′, B″, C′ and C″. Images in this figure are representative of five repeats for the $arm^{YD35}$ and $shg^{R69}$ experiments, and four repeats for the $\alpha\text{-}cat^1$ experiments. Scale bars: 10 μm.

depleted of E-cadherin ($shg^{R69}$) (11.9±3.6 μm, mean±s.d.) compared to the neighbouring heterozygous cells (7.8±2.7 μm) (Fig. 4A,D). Similarly, enterocytes without β-catenin had longer septate junctions (10.7±4.6 μm) relative to those in control cells (6.6±1.9 μm) (Fig. 4B,D). Cells lacking α-catenin also had slightly longer septate junctions (7.1±3.0 μm versus 6.1±2.7 μm), but this difference was not significant (Fig. 4C,D). Thus, lateral E-cadherin adhesion complexes limit the basal extent of the septate junctions. Consistent with this, the domain of the lateral membrane below the septate junctions (the sub-septate junction domain) was significantly shorter in both $shg^{R69}/shg^{R69}$ enterocytes (9.6±3.6 μm versus 18.2±4.1 μm) and in $arm^{YD35}/arm^{YD35}$ enterocytes (12.4±5.1 μm versus 17.9±5.6 μm) (Fig. 4A,B; Fig. S4E). This decrease in the sub-septate junction domain was not observed upon loss of function of α-catenin (15.0±3.7 μm versus 15.2±5.5 μm) (Fig. 4C; Fig. S4E). The strong reduction in the sub-septate junction domain in $shg^{R69}/shg^{R69}$ enterocytes more than compensated for the longer septate junctions in these cells, leading to a significant decrease in the overall length of the lateral domain (21.5±5.4 μm) compared to that in neighbouring heterozygous cells (26.0±4.6 μm) (Fig. 4A,E). The reduction in total lateral domain length was not significant in enterocytes that lacked β-catenin or α-catenin, consistent with their weaker effects on the size of the sub-septate junction domain (Fig. 4B,C,E). To confirm that the reduction in the length of the sub-septate junction domain and the concomitant basal expansion of the septate junctions was caused by the mutations in *shotgun* and *armadillo*, and not by second hits on the mutant chromosomes, we repeated these experiments with another null allele of each gene ($shg^{IG29}$ and $arm^{XP33}$). Consistent with our previous results, both $shg^{IG29}$ and $arm^{XP33}$ mutant cells showed a significant increase in the extent of the septate junction domain and a decrease in the length of the lateral membrane below the septate junctions (Fig. S4A–D). However, only $shg^{IG29}$ mutant cells showed a significant reduction in total cell height, indicating that the loss of E-cadherin has a stronger effect on the lateral domain organisation than loss of either catenin.

To investigate why septate junctions elongate in the absence of E-cadherin, we measured the fluorescence intensity of septate junction proteins in $shg^{R69}$, $arm^{YD35}$ and $\alpha\text{-}cat^1$ mutant clones. In all cases, the concentration of septate junction proteins was similar to that in the respective control enterocytes (Fig. S4F). This indicates that the elongation of the septate junctions is not simply the result of the passive lateral diffusion of their components in the membrane, as this would reduce their concentration at each point. Thus, enterocytes lacking E-cadherin and β-catenin localise significantly more protein along their septate junction domains relative to control enterocytes (Fig. 4F). Altogether this suggests that the absence of E-cadherin-based adhesions elicits a compensatory mechanism that expands the septate junction domain by localising more septate junction proteins.

### N-cadherin is not redundant with E-cadherin

One possible reason for the lack of a clear epithelial polarity phenotype in E-cadherin null cells is redundancy with N-cadherin, as is the case in the follicular epithelium that surrounds developing *Drosophila* egg chambers (Straub et al., 2011; Tanentzapf et al., 2000). Indeed, single-cell RNA sequencing suggests that N-cadherin is weakly expressed in the midgut (Hung et al., 2020; Li et al., 2022). To test this directly, we examined the expression of N-cadherin endogenously tagged with GFP (Chan et al., 2017). Ncad::GFP was not detectable in the enterocytes under normal conditions, but was clearly expressed in the ISCs (Fig. 5A, left-hand panel). After knockdown of E-cadherin using $shg^{RNAi}$, however, Ncad::GFP was also visible at the lateral membrane of the enterocytes (Fig. 5A, right-hand panel). This suggests that N-cadherin is upregulated in the absence of E-cadherin and might compensate for the absence of the latter.

If N-cadherin can compensate for the loss of E-cadherin in the enterocytes, the concomitant depletion of both cadherins should produce a stronger phenotype than removing either alone. We therefore generated cells lacking both cadherins by inducing MARCM clones homozygous for a null allele of E-cadherin ($shg^{IG29}$) in which N-cadherin was also knocked down by RNAi. These GFP-positive cells lacking both cadherins from birth still integrated into the epithelium and displayed normal apical-basal polarity (Fig. 5B,C). Depleting N-cadherin from E-cadherin mutant clones partially suppressed the extension of the septate junctions observed in $shg^{IG29}$ homozygous clones but did not rescue the reduction of the length of the entire lateral domain induced by

Journal of Cell Science

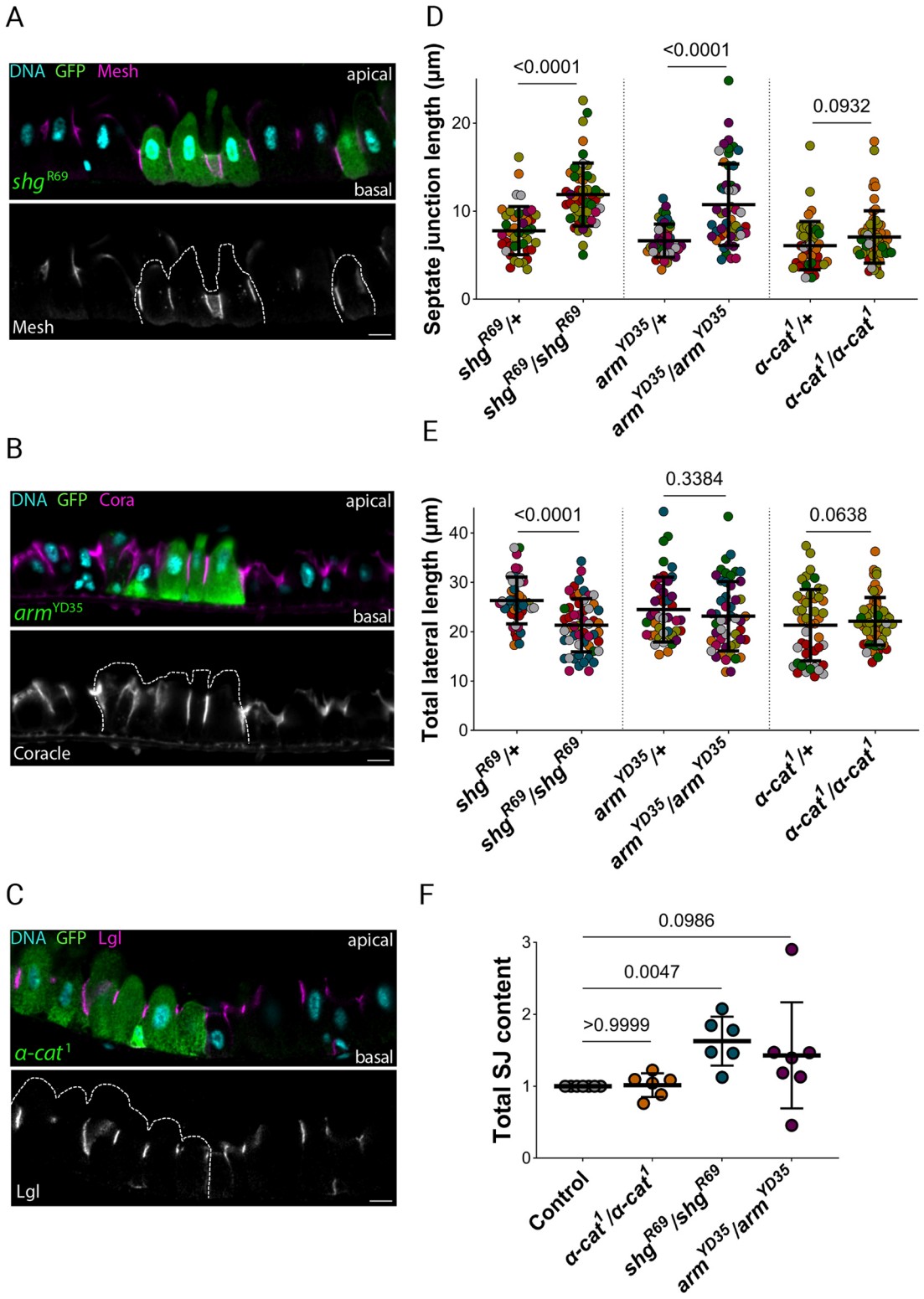

**Fig. 4. Loss of E-cadherin and β-catenin extends the septate junction domain.** (A) An apical-basal section through a midgut containing a *shg*[R69] homozygous clone marked by GFP expression (green) stained for DNA (DAPI, blue) and Mesh (magenta). (B) An *arm*[YD35] homozygous clone (GFP; green) stained for DNA (DAPI; blue) and Coracle (magenta). (C) An *α-cat*[1] homozygous clone (GFP; green) stained for DNA (DAPI; blue) and Lgl (magenta). In A–C, the mutant cells are outlined with white dashed lines in the lower panel. (D) A superplot showing septate junction length in *shg*[R69]/+ (6 midguts, 39 junctions), *shg*[R69]/*shg*[R69] (6 midguts, 46 junctions), *arm*[YD35]/+ (7 midguts, 44 junctions), *arm*[YD35]/*arm*[YD35] (7 midguts, 50 junctions), *α-cat*[1]/+ (4 midguts, 47 junctions) and *α-cat*[1]/*α-cat*[1] (4 midguts, 61 junctions) enterocytes. (E) A superplot showing the total length of the lateral domain in the same cells as in D. (F) A plot showing the intensity of multiple septate junction proteins (Mesh, Scrib, Coracle, Tsp2A and Lgl) along the entire length of the septate junctions in control (standardised mean=1, 7 midguts, 44 junctions), *arm*[YD35]/*arm*[YD35] (mean=1.43±0.74, 7 midguts, 50 junctions), *shg*[R69]/*shg*[R69] (mean=1.63±0.34, 6 midguts, 46 junctions) and *α-cat*[1]/*α-cat*[1] enterocytes (mean=1.02±0.17, 6 midguts, 74 junctions). Each circle represents the measurement for a single gut. Error bars in D–F are mean±s.d. *P*-values shown calculated with a Shapiro–Wilk normality test followed by the Kolmogorov–Smirnov test. Scale bars: 10 µm.

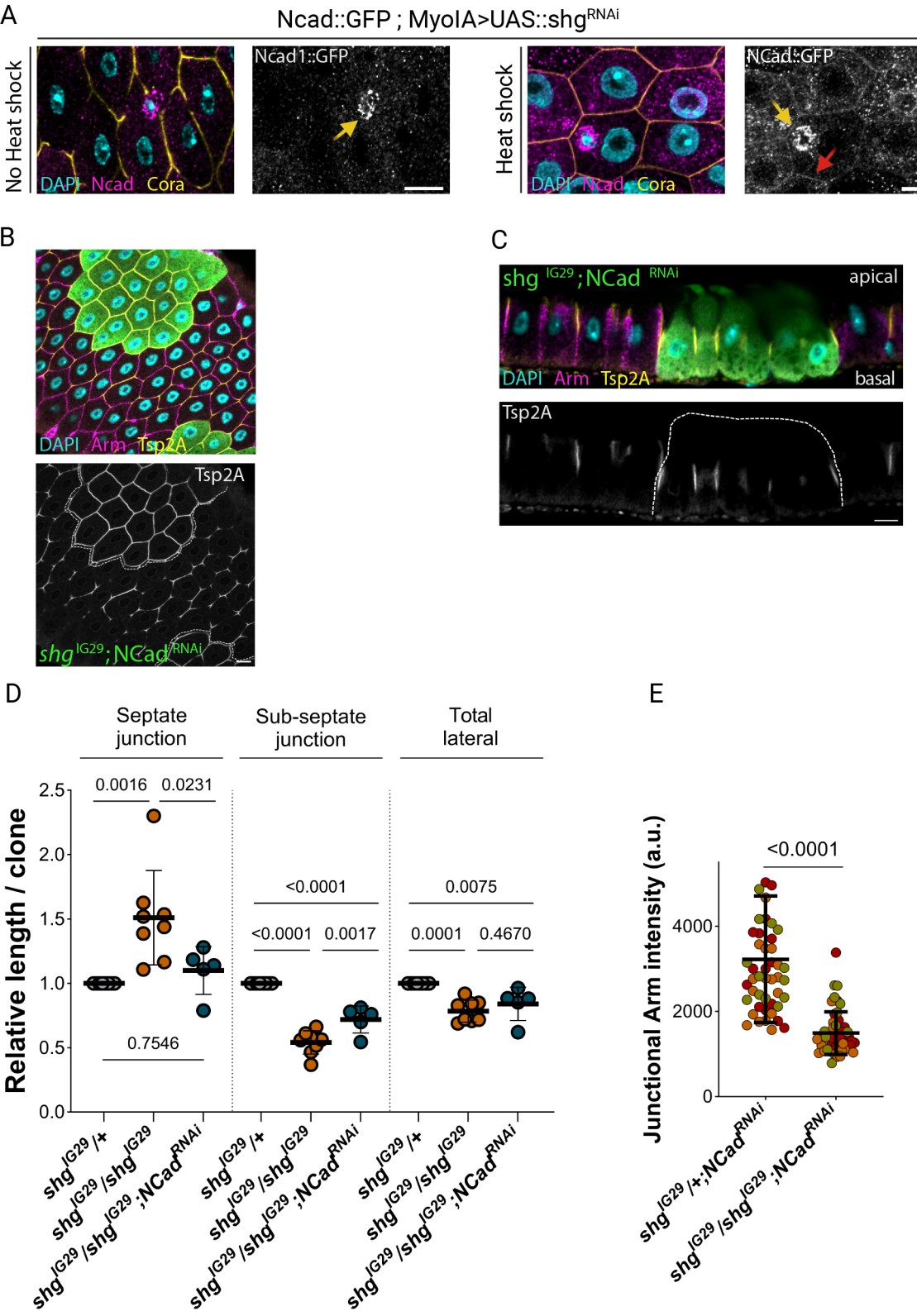

**Fig. 5.** See next page for legend.

E-cadherin loss (Fig. 5C,D). Upon loss of E- and N-cadherin, β-catenin no longer localised to the sub-septate junction domain, as expected (Fig. 5B,C,E). Thus, N-cadherin does not compensate for the loss of E-cadherin, and both cadherins are dispensable for epithelial polarity and organisation in the midgut.

**Cadherin-mediated adhesion is dispensable for epithelial polarity**

In addition to E- and N-cadherin, the *Drosophila* genome encodes for an additional classical cadherin, CadN2, which lies a few kilobases 3′ to N-cadherin. Like N-cadherin, CadN2 is weakly

**Fig. 5. N-cadherin 1 does not compensate for loss of E-cadherin in the midgut epithelium.** (A) Confocal sections through midguts endogenously expressing N-cadherin 1::GFP (magenta), without *shotgun* RNAi expression (no heat shock, left-hand panel) or with *shotgun* RNAi expression (heat shock, right-hand panel) stained for DNA (DAPI; blue) and Coracle (yellow); the yellow arrows indicate ISCs and the red arrow marks a lateral junction. (B) A horizontal confocal section through the basal region of a midgut containing a $shg^{IG29}$ mutant clone expressing $NCad1^{RNAi}$ (GFP; green), stained for DNA (DAPI; blue), Tsp2A (yellow) and Armadillo (magenta); the clone is outlined with a white dashed line. (C) An apical-basal section through a midgut containing a $shg^{IG29}$ mutant clone expressing $NCad1^{RNAi}$ (GFP; green), stained for DNA (DAPI; blue), Armadillo (magenta) and Tsp2A (yellow); the clone is outlined with a white dashed line. (D) Superplots showing the relative lengths of the septate junctions, the sub-septate junction domain and the whole lateral domain in $shg^{IG29}$ and $shg^{IG29}$; $NCad1^{RNAi}$ clones compared to heterozygous control cells in the same sample (Septate junction length: $shg^{IG29}$/+ standardised mean=1, 8 midguts; $shg^{IG29}$/$shg^{IG29}$ mean=1.51±0.37, 8 midguts; $shg^{IG29}$/$shg^{IG29}$; $NCad1^{RNAi}$ mean=1.10±0.19, 5 midguts; sub-septate junction domain length: $shg^{IG29}$/+ length standardized mean=1, 8 midguts; $shg^{IG29}$/$shg^{IG29}$ mean=0.54±0.09, 8 midguts; $shg^{IG29}$/$shg^{IG29}$; $NCad1^{RNAi}$ mean=0.72±0.11, 5 midguts); total lateral length ($shg^{IG29}$/+ standardised mean=1, 8 midguts; $shg^{IG29}$/$shg^{IG29}$ mean=0.78±0.09, 8 midguts; $shg^{IG29}$/$shg^{IG29}$, $NCad1^{RNAi}$ mean=0.84±0.13, 5 midguts). $shg^{IG29}$/+ and $shg^{IG29}$/$shg^{IG29}$ as in Fig. S4B; $shg^{IG29}$/+, $NCad1^{RNAi}$: 47 junctions; $shg^{IG29}$/$shg^{IG29}$; $NCad1^{RNAi}$: 53 junctions); Each circle in the plot shows the mean intensity for a single gut. (E) Measurements of Armadillo intensity at the junctions (sub-septate junction domain, $shg^{IG29}$/+, $NCad1^{RNAi}$: 48 junctions, 3 midguts; $shg^{IG29}$/$shg^{IG29}$, $NCad1^{RNAi}$: 52 junctions, 3 midguts) of $shg^{IG29}$ heterozygous and homozygous enterocytes depleted for N-cadherin. Error bars in D,E are mean±s.d. *P*-values shown calculated with a Shapiro–Wilk normality test followed by an unpaired *t*-test with Welch's correction. Scale bars: 10 µm.

expressed in enterocytes, and could potentially compensate for the loss of both E-cadherin and N-cadherin (Hung et al., 2020; Li et al., 2022). To investigate this possibility, we took advantage of the $Ncad^{\Delta14}$ allele, which is null for both N-cadherin and CadN2 proteins (Prakash et al., 2005). We generated $Ncad^{\Delta14}$ MARCM clones lacking both N-cadherin and CadN2, while simultaneously knocking down E-cadherin with $shg^{RNAi}$. Similar to our previous observations, enterocytes depleted of all three cadherins fully integrated into the epithelium and displayed polarised apical, lateral and basal domains (Fig. 6A,B). This indicates that the polarisation and integrity of the *Drosophila* gut epithelium does not require any of the classical cadherins.

Interestingly, we recovered far fewer $Ncad^{\Delta14}$; $shg^{RNAi}$ mutant clones than clones for $Ncad^{\Delta14}$ or $shg^{RNAi}$ alone (Fig. 6A,B; Fig. S5A,B). To quantify this effect, we scored the size and number of clones per gut of various genotypes at different time points after clone induction. When clones were induced by larval heat shock, $Ncad^{\Delta14}$ on its own gave a much lower frequency and smaller clones than $Ncad^{omb405}$, a null allele that only affects N-cadherin (Fig. 6C–C'''',D; Fig. S5B). By contrast $shg^{RNAi}$-expressing clones were found at high frequency and usually contained more than five labelled enterocytes (Fig. 6C'''',D; Fig. S5A). Knocking down E-cadherin in $Ncad^{\Delta14}$ clones with $shg^{RNAi}$ exacerbated the $Ncad^{\Delta14}$ phenotype – most guts had no detectable clones and the few that were recovered contained fewer than three labelled enterocytes (Fig. 6C–C'',D). This enhancement of the $Ncad^{\Delta14}$ 'small or no clone' phenotype on depletion of E-cadherin was not observed with other *Ncad* alleles ($Ncad^{B11}$ and $Ncad^{M19}$), indicating that it is specific for cells that lack both N-cadherin and CadN2 cadherins. Thus, removing all three cadherins strongly effects ISC or enterocyte survival or the development of enterocytes from mutant ISCs. We repeated these experiments using adult heat shock to induce

clones after the adult midgut had formed to rule out the possibility that the mutant cells were being lost during pupation. As observed with larval clone induction, more than 40% of the midguts contained no $Ncad^{\Delta14}$ clones and the few that occurred contained less than three cells. This phenotype was significantly enhanced when $shg^{RNAi}$ was also expressed to deplete E-cadherin, with more than 70% of midguts lacking any detectable clones (Fig. 6C–C'',E). Altogether, these data indicate that although the cadherins are dispensable for polarity, they are important for the survival and/or differentiation of the ISCs and/or their progeny.

## DISCUSSION

Our results demonstrate that adherens junctions are not required for the organisation or polarisation of the midgut epithelium. Not only do single mutants in E-cadherin, β-catenin and α-catenin have no effect on apical-basal polarity, but the enterocytes still polarise normally after the simultaneous removal of all classical cadherins – E-cadherin, N-cadherin and CadN2. This contrasts with observations in other epithelia, where the loss of E-cadherin adhesion complexes disrupts epithelial organisation to produce mesenchymal-like cells with disrupted polarity (Bonello et al., 2021; Capaldo and Macara, 2006; Cox et al., 1996; Larue et al., 1994; van Roy and Berx, 2008; Sheppard et al., 2022; Tunggal et al., 2005). This reinforces the evidence that the midgut epithelium polarises by a different mechanism from other *Drosophila* epithelia. Indeed, whereas ectodermal epithelia can polarise in the absence of septate junctions, which form late in epithelial differentiation, the septate junctions are essential for apical domain formation in the midgut (Chen et al., 2018; Laprise et al., 2009; Tepass et al., 2001). The septate junctions in the midgut form in the position that the adherens junctions occupy in other fly epithelia. Thus, it might be that the key step in apical-basal polarisation is the formation of a junction at the boundary between the apical and lateral domains, with the septate junctions taking over this role from the adherens junctions in the midgut.

Although E-cadherin adhesion complexes are dispensable for polarity in the midgut, they do play a role in determining the relative sizes of the septate junction and sub-septate junction domains. Not much is known about how the sizes of membrane domains are controlled, but this presumably (1) depends on the rates at which exocytosis delivers membrane and membrane proteins to the domain, (2) the rate at which endocytosis removes them, and (3) the factors that define the boundaries between domains. The reduction in the size of the sub-septate junction domain is likely to be due at least in part to the failure to deliver exocytic vesicles containing E-cadherin. It is also possible that E-cadherin promotes the delivery of other proteins to this domain either by co-transport or by enhancing the delivery of other exocytic vesicles as observed in mammalian cells (Grindstaff et al., 1998; Yeaman et al., 2004). The exocyst associates with β-catenin and localises to adherens junctions in the *Drosophila* pupal notum (Langevin et al., 2005). Thus, E-cadherin–β-catenin complexes could enhance secretion to the sub-septate junction domain by recruiting the exocyst, which then captures exocytic vesicles and tethers them to the plasma membrane prior to their fusion (Polgar and Fogelgren, 2018). E-cadherin could also maintain the size of the sub-septate junction domain by organising the sub-membrane F-actin scaffold to anchor membrane proteins and antagonise their endocytosis. This seems less likely, however, given that loss of α-catenin, which links E-cadherin adhesion complexes to F-actin, does not lead to a decrease in the size of the domain (Desai et al., 2013).

The expansion of the septate junctions in *shg* and *arm* mutants suggests that E-cadherin adhesion complexes somehow limit their basal extension. This could be through these complexes

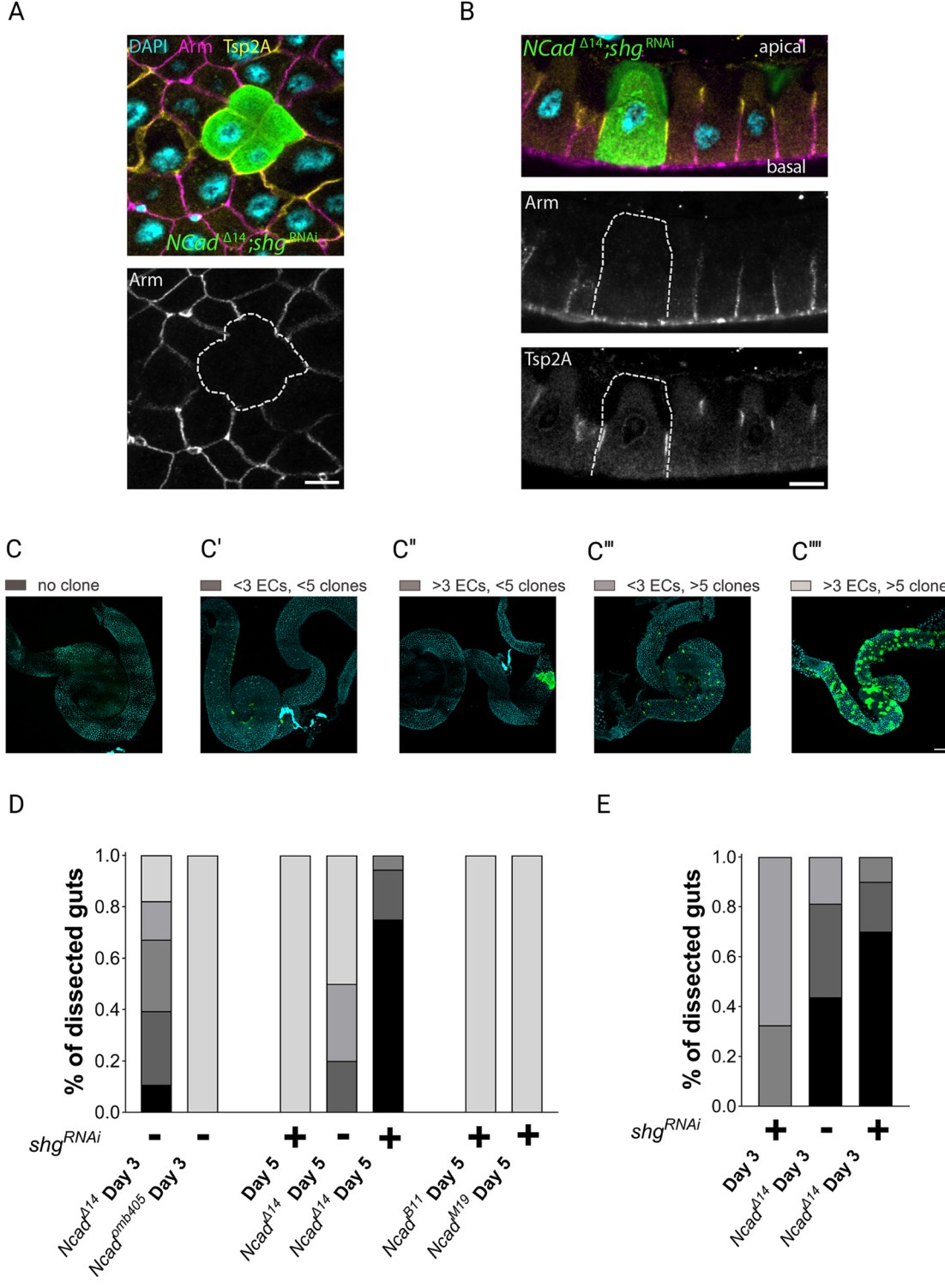

**Fig. 6. All classical cadherins are dispensable for apical-basal polarisation of the midgut epithelium**. (A) A horizontal view of a midgut containing a $Ncad^{\Delta 14}$ mutant clone expressing $shg^{RNAi}$ (GFP; green), stained for DNA (DAPI; blue), Tsp2A (yellow) and Armadillo (magenta); the clone is outlined with a white dashed line in the lower panel. Scale bar: 10 µm. (B) An apical-basal section through a midgut containing a $Ncad^{\Delta 14}$ mutant clone expressing $shg^{RNAi}$ (GFP; green), stained for DNA (DAPI; blue), Armadillo (magenta) and Tsp2A (yellow). Scale bar: 10 µm. Images in A,B are representative of 12 midguts from four experiments. (C) Analysis of the frequency and size of $Ncad^{\Delta 14}$, $shg^{RNAi}$ and $Ncad^{\Delta 14}$, $shg^{RNAi}$ clones after larval (L.H.S.) and adult heat shock (A.H.S.). Representative images of guts in the different categories of clone frequency and size are shown. C, no clones; C′, fewer than three enterocytes (ECs) per clone and fewer than five clones per gut; C″, more than three ECs per clone but fewer than five clones per gut; C‴, fewer than three ECs per clone with more than five clones per gut; C‴′, more than three ECs per clone and more than five clones per gut. Scale bar: 100 µm. (D) Bar chart showing the percentage of guts in each category of clone size and frequency after larval heat shock ($Ncad^{\Delta 14}$ Day 3 L.H.S., 27 midguts; $Ncad^{omb405}$ Day 3 L.H.S., 35 midguts; $shg^{RNAi}$ Day 5 L.H.S., 8 midguts; $Ncad^{\Delta 14}$ Day 5 L.H.S., 10 midguts; $Ncad^{\Delta 14}$, $shg^{RNAi}$ Day 5 L.H.S., 22 midguts; $Ncad^{B11}$, $shg^{RNAi}$ Day 5 L.H.S., 2 midguts; $Ncad^{M19}$, $shg^{RNAi}$ Day 5 L.H.S., 21 midguts) and adult heat shock ($shg^{RNAi}$ Day 3 A.H.S., 18 midguts; $Ncad^{\Delta 14}$ Day 3 A.H.S., 16 midguts; $Ncad^{\Delta 14}$, $shg^{RNAi}$ Day 3 A.H.S., 11 midguts) in each mutant.

acting as a barrier that prevents septate junction proteins spreading more basally or they could promote septate junction protein endocytosis at the boundary between the two domains. A third possibility is that septate junction proteins are upregulated in *shg* and *arm* mutants, leading to lengthening of the septate junctions to compensate for the absence of Cadherin adhesion complexes. As is the case for the sub-septate junction domain, loss of α-catenin has no significant effect on the size of the septate junctions, making it unlikely that the linkage of E-cadherin to F-actin plays a role in establishing the border between the adherens and septate junctions.

Neither N-cadherin nor CadN2 appear to play any significant role in the formation of junctions in the enterocytes under normal conditions, but N-cadherin is upregulated upon knockdown of E-cadherin and localises to adherens junctions. Surprisingly, the depletion of N-cadherin partially suppresses the expansion of the septate junction domain seen upon loss of E-cadherin alone, for reasons that are unclear. Furthermore, N-cadherin does not appear to recruit β-catenin to cell–cell adhesions, given that β-catenin does not detectably localise to the membrane in the sub-septate junction domain in the absence of E-cadherin, even though N-cadherin is expressed and localised under these conditions.

Although the classical cadherins are not required for enterocyte polarity, they do affect cell survival or the development of ISCs into mature enterocytes, as we recovered almost no enterocyte-containing clones that lacked E-cadherin, N-cadherin and CadN2. N-cadherin is strongly expressed in ISCs even when E-cadherin is present. This suggests that the cadherins function redundantly with each other in the stem cells and that the almost complete failure to recover clones depleted for all three cadherins is because they are required for ISC survival, division or maintenance. An alternative possibility is that they function in cell–cell adhesion between ISCs and EBs. E-cadherin mediates adhesion between the two daughters of an ISC division, which maintains contact between the cells while Delta–Notch signalling induces one daughter cell to adopt the EB fate (Maeda et al., 2008). It is therefore possible that all three cadherins mediate ISC–EB adhesion and, in their complete absence, Delta–Notch signalling does not usually occur. This would lead to a failure to specify EBs and could account for the lack of enterocyte-containing triple knockdown clones.

## MATERIALS AND METHODS
### Drosophila stocks
The following *Drosophila melanogaster* stocks were used in this study: $w^{1118}$, *FRT42D shg*$^{R69}$ (Godt and Tepass, 1998); *FRTG13 shg*$^{IG29}$ (Tepass et al., 1996); *w; FRTG13* [Bloomington *Drosophila* Stock Center (BDSC), 1956]; *FRTG13 shg*$^1$ (BDSC, 58471); *hs–Flp FRT40A; da–Gal4 UAS–mCD8::GFP α-Cat*$^1$/*TM6B* (Sarpal et al., 2012); *tub–Gal80 ubi–α-Cat FRT40A; act5c–Gal4 α-Cat*$^1$/*TM6b, FRT19A arm*$^{YD35}$ (Riggleman et al., 1989), *FRT19A arm*$^{XP33}$ (Riggleman et al., 1989); *Ncad::GFP* (Chan et al., 2017); *FRT40A Ncad*$^{Δ14}$ (Prakash et al., 2005); *FRT40A Ncad*$^{omb405}$ and *FRT40A Ncad*$^{B11}$ (Lee et al., 2001); *FRT40A Ncad*$^{M19}$ (Iwai et al., 1997); *y w, UAS-mCD8::GFP, Act5C-GAL4, hsFLP[1]; FRTG13 tubP-GAL80* (Lee and Luo, 1999); *hsFlp; FRTG13 tub-Gal80; tub-Gal4, UAS-mCD8-GFP* (generated by Dr Jia Chen, The Gurdon Institute, University of Cambridge, UK); *y w, UAS-mCD8::GFP, Act5C-GAL4, hsFLP; FRT40A tubP-GAL80* (Chen et al., 2018); *w, hsFLP, tubP-GAL80, FRT19A;; tubP-GAL4, UAS-mCD8::GFP/TM3, Sb* (Lee and Luo, 1999), *y w hsFLP tub-GAL4 UAS-GFPnls/FM7; FRT42D tub-GAL80/CyO* (Caygill and Brand, 2017), *UAS-shg-RNAi* (BDSC, 32904), *UAS-shg-RNAi* (BDSC, 32428), *UAS-arm-RNAi* (BDSC, 35004), *UAS-α-cat-RNAi* (BDSC, 33430), *UAS-NCad-RNAi* (BDSC, 41982), *UAS-NCad-RNAi* (BDSC, 27503), *y w; MyoIA-GAL4, tubP-GAL80*$^{ts}$ (gift from Golnar Kolahgar, Department of Physiology, Development and Neuroscience, University of Cambridge, UK).

### Drosophila maintenance, crosses and genetics
*D. melanogaster* stocks were maintained on standard medium (https://studylib.net/doc/7322751/iberian-fly-food) supplemented with dry yeast at 18°C or room temperature (∼22°C) prior to experimental procedures. We used the MARCM technique to generate mutant clones marked by GFP expression within otherwise heterozygous midguts (Lee and Luo, 1999). Crosses were grown at 25°C for 7 (prepupa stage) or 13 days (mated 3-day-old adults) and subsequently subjected to heat shock. Heat shock was performed at 37°C for 1 h, twice per day, for 2 to 5 days. Flies were frequently transferred to new vials to decrease population density and exchange the medium. After heat shock, flies were transferred to 25°C until dissection. The female flies used in the experiments were always kept with male siblings and their guts dissected 3 to 16 days after the last heat shock. When expressing RNAi constructs in clones using the MARCM technique, crosses were maintained at 18°C and placed at 29°C after heat shock until dissection. For RNAi experiments driven by MyoIA-GAL4, crosses were grown at 18°C, and 3-day-old adults were transferred to 29°C for 7–12 days before dissection.

### Immunofluorescence and microscopy
Midgut dissections and immunostainings were performed as described in Chen et al. (2018) with the following modifications. Female midguts were dissected in a depression slide containing 1× PBS for a maximum of 5 min. Dissected midguts were transferred to a 3–10 cm plastic cylinder sealed with a wire mesh and submerged in 1× TSS solution (0.03% Triton X-100, 4 g l$^{-1}$ NaCl; 95°C) at 78°C for 14 s followed by 1 min in 1× TSS solution on ice. Heat-fixed guts were then transferred to an Eppendorf tube containing 8% PFA in 1× PBST (0.1% Triton X-100 in PBS) and fixed for 20 min at room temperature. Guts were washed three times (10 min each) with 1×PBST and subsequently blocked with 2% normal goat serum (NGS; Stratech Scientific Ltd, cat. #005-000-121; concentration of stock solution: 10 mg/ml) in 1× PBST for 1 h at room temperature. Midguts were incubated with primary antibodies in 2% NGS in 1× PBST overnight at 4°C. The primary antibodies used in this study were: anti-Arm [Developmental Studies Hybridoma Bank (DSHB), N2 7A1, mouse 1:100], anti-Mesh (gift from Mikio Furuse, National Institute for Physiological Sciences, Okazaki, Japan, rat 1:1000), anti-Scrib (gift from Chris Q. Doe, Institute of Neuroscience, University of Oregon, USA, rabbit 1:1000), anti-Canoe (gift from Mark Peifer, Dept. of Biology, University of North Carolina, USA, rabbit 1:1000), anti-Tsp2A (gift from Mikio Furuse, rabbit 1:1000), anti-Coracle (DSHB, C615.16, mouse 1:100), anti-Lgl (Santa Cruz Biotechnology, SC98260, rabbit 1:250), anti-aPKC (Santa Cruz Biotechnology, SC216, rabbit 1:100), anti-Nrv (DSHB, nrv5F7, mouse 1:100) and anti-GFP (AbCam, #ab13970, chicken, 1:1000). Midguts were washed three times for 20 min each time, at room temperature with 1× PBST before incubation with secondary antibodies in 2% NGS in 1× PBST for 2 h at room temperature. The secondary antibodies used in this study were: Alexa Fluor 488 conjugated anti-mouse IgG (#A11029, 1:1000), Alexa Fluor 488-conjugated anti-rabbit IgG (#A11034, 1:1000), Alexa Fluor 488-conjugated anti-chicken IgY (#A11039, 1:1000), Alexa Fluor 555-conjugated anti-rat IgG (#A21434, 1:1000), Alexa Fluor 555 anti-mouse IgG (#A21422, 1:1000), Alexa Fluor 555-conjugated anti-rabbit IgG (#A21428, 1:1000), Alexa Fluor 647 anti-mouse IgG (#A21236, 1:500), Alexa Fluor 647-conjugated anti-rabbit IgG (#A21245, 1:500) and Alexa Fluor 647-conjugated anti-rat IgG (#A21247, 1:500). After incubation with secondary antibodies, the midguts were washed three times for 20 min each time, with 1× PBST and afterwards mounted in Vectashield containing DAPI (Vector Laboratories). For Ncad1::GFP staining, guts were dissected in PBS and fixed for 1 h at room temperature in methanol-free 4% paraformaldehyde (Thermo Fisher Scientific, 28908). Subsequently, guts were washed twice for 30 min in PBST, then incubated in anti-GFP primary antibody (as above) diluted in PBST (1:1000) for 48 h at 4°C. The next day, guts were washed twice (30 min each) in PBST and incubated in secondary antibodies and phalloidin conjugated to Alexa Fluor 647 (Thermo Fisher Scientific, A22287, 1:500) or Phalloidin-iFluor 405 (Abcam 176752, 1:500) diluted in PBST overnight at 4°C. Finally, guts were washed twice in PBST (30 min each) and mounted on a glass slide with Vectashield containing DAPI (Vector Laboratories).

Images were acquired on either an Olympus IX81 confocal microscope using FluoView FV1000 Laser Scanning software (4.2.1.20) and a 60× oil objective (UPLSAPO, NA=1.35) or a Zeiss 880 Airyscan microscope using the confocal mode, using the Plan-Apochromat 63× oil objective (1.4 NA). Z-stacks of guts were acquired from the top until the middle section to reveal the longitudinal profile of the enterocytes at an 800×800 size (pixel size=0.265 μm) with 1 μm step size between frames with a 12-bit depth. Images were processed using ImageJ (Schindelin et al., 2012).

## Quantification of enterocyte length and circularity

The ImageJ 'segmented line tool' plugin was used to measure the length of the lateral domain. The 'total lateral length' corresponds to the distance between the start of the septate junctions on the apical side until the basement membrane of enterocyte cells (excluding the muscle layer). The 'septate junction length' corresponds to the total length of the given septate junction marker, whereas the 'sub-septate junction length' corresponds to the difference between the 'septate junction length' and the 'total lateral length'. Each measurement was carried out in the most longitudinal view of each septate junction marker in cells located on the middle section of the gut. To quantify enterocyte circularity, we used the polygon tool in ImageJ, where the basal area of each enterocyte cell was manually delineated using either the GFP from the MARCM system or lateral membrane marker staining. The value of 1 represents a perfect circle. These analyses were carried out by a researcher that was aware of the experimental conditions.

## Quantification of fluorescence intensity

To plot Arm intensities across the adherens junction in Fig. 2, a straight line was drawn between two nuclei centred at the AJs. To quantify Armadillo intensity at the level of the adherens junctions, we used the segmented line tool with a width of three. In heterozygous enterocytes, the adherens junctions were identified by Armadillo staining, whereas in the homozygous enterocytes, cell–cell borders are identified either by the GFP from the MARCM clones or by the relative position of a septate junction marker. Line segments were transferred to the Armadillo channel to measure the intensities. To quantify cytoplasmic Armadillo intensity, an oval shape of fixed size – equal for heterozygous and homozygous ECs in each image – was drawn between the lateral domain and the nucleus and measured in the Armadillo channel. To quantify septate junction protein intensities, we used the segmented line tool with a width of 3 along the entire length of each septate junction on its most longitudinal view. Integrated intensity values were used for graphic representations.

## Normalisation, statistics and experimental reproducibility

To compare measurements between clones of different mutants, the homozygous mutant individual cell measurements (length of the lateral domain, relative armadillo intensities and the intensity of the septate junction proteins) were first averaged, and the resulting mean divided by the average of the individual measurements in heterozygous enterocytes from the same sample. All controls (heterozygous enterocytes in each clone) were standardised to 1. SuperPlots were used in all graphic visualisations except when directly comparing clones that are represented by a single measurement (Lord et al., 2020). The Shapiro–Wilk test was used to assess normality in all distributions. The Kolmogorov–Smirnov test was used in Figs S1B,E,F, S3D, Fig. 4D, Fig. S4E ($shg^{R69}$ and $arm^{YD35}$), 4E ($\alpha$-cat$^1$ and $arm^{YD35}$), S4B ('septate junction length'), S4D ('septate junction length'), S4F ($\alpha$-cat$^1$ and $arm^{YD35}$); unpaired t-tests with Welch's correction were used in Figs S1C, S4E ($\alpha$-cat$^1$), S4F ($shg^{R69}$), 4E ($shg^{R69}$), S4B ('sub-septate junction length' and 'total lateral length'), S4D ('sub-septate junction length' and 'total lateral length') and 5E. The Kruskal–Wallis with Dunn's post hoc test was used in Figs 2G,H and 4F; a one-way ANOVA with Tukey's post hoc test was used in Fig. 5D. The mean±s.d. is represented in black in each distribution and statistical significance is taken as $P<0.05$. GraphPad prism (10.0.3) was used to build the graphics and to perform statistical analyses. The number of individual experiments or crosses per condition is as follows: UAS::shg-RNAi (32428), one experiment; UAS::shg-RNAi (32904), two experiments; UAS::arm-RNAi (35004), one experiment; UAS::$\alpha$-cat-RNAi (33430), two experiments; UAS::Ncad-RNAi (41982), one experiment; $\alpha$-cat$^1$, four experiments (10 midguts);

$arm^{YD35}$, five experiments (16 midguts); $arm^{XP33}$, two experiments (10 midguts); $shg^{IG29}$, five experiments (17 midguts); $shg^{R69}$, five experiments (18 midguts); $shg^{IG29}$, Ncad RNAi, three experiments (14 midguts).

## Acknowledgements

We would like to thank Mikio Furuse, Chris Doe, Mark Peifer and Ulrich Tepass for fly stocks and antibodies, Jia Chen for help and advice on midgut preparation and staining, the Gurdon Media team for fly food and the lab members of the D.S.J. lab for technical support.

## Competing interests

D.S.J. is a director of The Company of Biologists but was not included in any aspect of the editorial handling of this article or peer review process. The authors declare no other competing or financial interests.

## Author contributions

Conceptualization: D.S.J., C.A.C.; Data curation: C.A.C., M.A.T.; Formal analysis: C.A.C., M.A.T.; Funding acquisition: D.S.J.; Investigation: C.A.C., M.A.T.; Methodology: C.A.C., M.A.T.; Project administration: D.S.J.; Resources: D.S.J.; Supervision: D.S.J.; Validation: D.S.J.; Visualization: M.A.T., C.A.C., D.S.J.; Writing – original draft: D.S.J., C.A.C.; Writing – review & editing: D.S.J., C.A.C.

## Funding

This work was supported by a Wellcome Trust Principal Fellowship to D.S.J. (080007, 207496) and by core support from the Wellcome Trust (092096, 203144) and Cancer Research UK (A14492, A24823). C.A.C. was supported by a scholarship from the Fundação para a Ciência e Tecnologia, Portugal (PD/BD/52194/2013) and MAT by an EU Marie Skłowdowska Action postdoctoral fellowship (GA797837). Open Access funding provided by University of Cambridge. Deposited in PMC for immediate release.

## Data and resource availability

All relevant data can be found within the article and its supplementary information.

## Peer review history

The peer review history is available online at https://journals.biologists.com/jcs/lookup/doi/10.1242/jcs.263644.reviewer-comments.pdf

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
