## [Peer Review File · Journal of Cell Science]

Adherens junctions limit septate junction length in *Drosophila* midgut enterocytes but are not required for polarity

Cátia A. Carvalho, Mihoko A. Tame and Daniel St Johnston

DOI: 10.1242/jcs.263644

Editor: Richa Rikhy

Review timeline

Original submission: 22 March 2024

Editorial decision: 13 May 2024

First revision received: 2 May 2025

Accepted: 2 June 2025

Original submission

First decision letter

MS ID#: JOCES/2024/262144

MS TITLE: Adherens junctions limit the extent of the septate junctions in *Drosophila* midgut enterocytes but are not required for apical-basal polarity

AUTHORS: Catia A Carvalho, Mihoko Amy Tame, and Daniel St Johnston

ARTICLE TYPE: Research Article

Dear Dr. St Johnston,

We have now reached a decision on the above manuscript.

To see the reviewers' reports and a copy of this decision letter, please go to: <https://submit-jcs.biologists.org> and click on the 'Manuscripts with Decisions' queue in the Author Area. (Corresponding author only has access to reviews.)

As you will see, the reviewers raise some criticisms that prevent me from accepting the paper at this stage. However, a revised version might prove acceptable, if you can address their concerns. Please address each of the concerns raised by the reviewers. In particular, please address the concerns regarding the need for DE-cad and N-cad in this system. Please add a discussion of the need for septate junctions in mediating polarity based on previous findings. Please make changes in the introduction to signify the questions addressed in the study. If you think that you can deal satisfactorily with the criticisms on revision, I would be pleased to see a revised manuscript. We may then return it to the reviewers.

Reviewer 1 Advance Summary and Potential Significance to Field:

This manuscript is part of ongoing efforts by St. Johnston and his colleagues to understand the cellular mechanisms that mediate epithelial biogenesis in the fly midgut. Their earlier work revealed that this is an unusual system: its polarized organization of junctions in the apical junctional complex is unlike that of other *Drosophila* epithelia and more like vertebrate epithelia; cells in this system polarize in a basal-to-apical direction; nor do these seem to involve canonical polarity factors.

Now they ask whether the cadherin adhesion system is necessary for polarization in the midgut, as it has been implicated in other epithelia. Their data indicate that the answer is no: the midgut polarizes largely normally when the E-cadherin system is disrupted and, although the cells also express N-cadherin, this is not compensating since polarization continues when both cadherins are targeted. The maneuvers do affect cell shape and are associated with expansion of the septate junctions, but these are relatively minor. The dominant result is that polarization doesn't seem to require the classical cadherin system

Reviewer 1 Comments for the Author:

The analysis is logical and the data are high quality. These results are useful for the community. The weakness for me is that, in its current form, the manuscript is of rather specialist interest. It doesn't provide insight into how something as physiologically fundamental as epithelial polarity is being established without the input of "mainstream" mechanisms. It is not clear where such insight might come from, but possible directions include:

- A) The study uses genetic approaches that will create sustained protein loss, so is it possible that more acute inhibition will reveal effects that are then compensated for (albeit by unknown mechanisms)?
- B) Could the key to the midgut system lie in the septate junctions? Is polarity affected if septate junctions are perturbed (or if septate junctions are perturbed in E-cadherin mutant animals)?

Reviewer 2 Advance Summary and Potential Significance to Field:

This study thoroughly explores the requirement for adherens junction components in the midgut enterocytes of adult *Drosophila*. This epithelium is of particular interest for its unusual polarity mechanism relative to the rest of *Drosophila* epithelia, bearing similarity to the organization of junctions in vertebrates. The authors find that AJ mutant cells in the midgut, unlike in other studied epithelia, are still able to polarize and integrate seemingly normally. However, in the absence of a functional AJ, the SJ domain is found to expand. These findings are valuable to the polarity field, with implications for domain formation, and the broader cell biology community.

Reviewer 2 Comments for the Author:

My comments and suggestions are below.

- In Figure 1, the authors may wish to consider recoloring the color palette to (1) match the rest of the paper, and (2) be color-blind friendly.
- In Figure S3, the images of mutant clones appear to contain different numbers of cells across the different AJ mutants. Is there an effect on clone size apart from enterocyte shape? It would be of interest to include cell numbers per clone, or at least a comment in the text.
- The use of different colors of data points in plots doesn't appear to be explained (e.g. Fig.4D-F and others). I assume the colors represent different guts, but this should be stated in the figure legend along with the number of guts for each sample.
- In Figure 4, the authors observe a significant increase in the length of SJs. Could the authors shed light on whether the increased length is due to existing proteins extending their domain, or an increased expression of SJ proteins? Including a measurement of SJ intensity might be one approach.
- The descriptive finding that polarity is maintained in gut enterocytes following E-cad RNAi was also reported in PMID 28847000 (Extended Data Fig 4); although the result was not quantitatively and thoroughly characterized as in this study, the authors may wish to consider citing it.
- In Figure 2A,C (and other figures), the use of the word 'BASAL' to describe the view from the bottom-up could be confusing (for example, SJ proteins are shown from this view), and the term is often used when apical-basal views are shown as in Figure 1. I might suggest leaving 'BASAL' out of the figures showing a bottom-up view, and rather simply describing it in the legend.
- In Figure 6, the wording of "Ncadi" could be confusing. The authors may wish to use a more standard indication that RNAi is used, such as with superscript ("Ncad RNAi" or "Ncad KD", for example).

Reviewer 3 Advance Summary and Potential Significance to Field:

The manuscript by Carvalho et al. describes the drosophila midgut enterocytes as a unique system where, unlike the other fly epithelia, adherens junctions are localized below the septate junctions on the lateral cell membranes, which resembles the arrangement in the vertebrate epithelia where adherens junctions are important in organizing polarity complexes. The current work follows up on the previous findings from the lab (Chen et al. 2018), that midgut enterocytes lack the expression of key polarity molecules such as - Bazooka (Par-3), and they maintain their polarity independent of the canonical polarity system, thus asking how is E-cadherins still utilized in this tissue to establish polarity. Using genetics perturbations and spatiotemporal localization, they have shown that E-cadherins depletion in the epithelia results in basal extension of the septate junction boundary, reduction of the sub-septate junction lateral membrane domain, and allows the cells to become more circular, without affecting polarity. Further, they show that N-cadherin does not compensate for E-cadherin's loss of function.

Altogether, this study indicates that the adherens junction regulates septate junction length but not apicobasal polarity in this endodermal origin epithelial system.

While the manuscript is written clearly, it was hard for me to understand what the key question is and what are the important takeaways from this study. In the current format, it is a series of interesting observations without providing insight into the underlying mechanisms and why these observations matter for tissue morphogenesis. The results and discussion section can be further refined to discuss the current findings in the context of what's known and what it contributes towards further advancing the field. Further, the lab has already studied and established that the integrin adhesion complex-septate junction axis determines EC polarity (Chen et al. 2018). Now they find adherens junctions are dispensable for midgut enterocytes polarity. However, how is this important for tissue development and morphogenesis is not clear.

Reviewer 3 Comments for the Author:

The manuscript by Carvalho et al. describes the drosophila midgut enterocytes as a unique system where, unlike the other fly epithelia, adherens junctions are localized below the septate junctions on the lateral cell membranes, which resembles the arrangement in the vertebrate epithelia where adherens junctions are important in organizing polarity complexes. The current work follows up on the previous findings from the lab (Chen et al. 2018), that midgut enterocytes lack the expression of key polarity molecules such as - Bazooka (Par-3), and they maintain their polarity independent of the canonical polarity system, thus asking how is E-cadherins still utilized in this tissue to establish polarity. Using genetics perturbations and spatiotemporal localization, they have shown that E-cadherins depletion in the epithelia results in basal extension of the septate junction boundary, reduction of the sub-septate junction lateral membrane domain, and allows the cells to become more circular, without affecting polarity. Further, they show that N-cadherin does not compensate for E-cadherin's loss of function.

Altogether, this study indicates that the adherens junction regulates septate junction length but not apicobasal polarity in this endodermal origin epithelial system.

While the manuscript is written clearly, it was hard for me to understand what the key question is and what are the important takeaways from this study. In the current format, it is a series of interesting observations without providing insight into the underlying mechanisms and why these observations matter for tissue morphogenesis. The results and discussion section can be further refined to discuss the current findings in the context of what's known and what it contributes towards further advancing the field. Further, the lab has already studied and established that the integrin adhesion complex-septate junction axis determines EC polarity (Chen et al. 2018). Now they find adherens junctions are dispensable for midgut enterocytes polarity. However, how is this important for tissue development and morphogenesis is not clear.

Major comments -

1. It is not clear from the manuscript why limiting the septate junction is essential for this tissue. E-cadherins knockdown resulting in extended septate junctions, alters cell shape/aspect ratio but doesn't affect the polarity of the ECs and it also does not seem to perturb gross tissue morphology. What is the functional consequence of these changes in cell shape or septate junction length on tissue integrity and morphogenesis?

2. What mechanism is underlying E-cadherin control of septate junction size?

Authors have evoked the role of exocytosis /endocytosis, but can they functionally test this?

3. An interesting observation in the manuscript suggests- that loss of N-cadherin can stabilize β -catenin, possibly because of compensatory E-cadherin upregulation. Another important finding in the manuscript is that N-cadherin can't compensate for E-cadherin. However, it is still not established whether N-cadherin has significant expression in this tissue, and it can be argued that the lack of expected response to N-cadherin knockdown is due to the lack of its expression. Thus, it would be important to show that N-cadherin is indeed expressed in the tissue and is downregulated upon RNAi treatment.

4. The authors find that upon loss of E- and N-cadherin, β -catenin accumulate in cytoplasm, eventually concluding that "N-cadherin plays a role in the destabilisation of β -catenin that is independent of E-cadherin
How did they reach to this conclusion? What is the experimental evidence? And why this is relevant for the tissue?

5. The first two figures are important descriptive results; however, I am finding it difficult to appreciate how they advance our knowledge. Also, these two figures can be easily combined into one figure.

6. Authors have only mentioned the number of cell-cell junctions for all the quantitative analysis plots. However, it will be useful to know the total number of animals used for quantification in each analysis.

First revision

Author response to reviewers' comments

Response to the referees comments

We thank the referees for their constructive comments and apologize for taking so long to revise our manuscript. The main reason for this delay was that we realized that the N-cadherin mutant that we had used only disrupted N-cadherin and left the adjacent *CadN2* gene intact. We have therefore repeated all our double knock out experiments using the *Ncad*^{D14} allele, which is null for both *Ncad* and *CadN2*. When we generate MARCM clones of *Ncad*^{D14} and also knock down E-cadherin with *shg* RNAi, we recover very few clones. Nevertheless, the clones we do obtain show normal apical-basal polarity. This confirms our conclusion that the midgut enterocytes are the only known epithelial cells that polarise in the complete absence of cadherins. The failure to recover large *Ncad*^{D14} *shg* RNAi clones may be due to a redundant requirement for any of the three cadherins for the survival, maintenance or proliferation of the ISCs.

Reviewer 1 Advance Summary and Potential Significance to Field:

This manuscript is part of ongoing efforts by St. Johnston and his colleagues to understand the cellular mechanisms that mediate epithelial biogenesis in the fly midgut. Their earlier work revealed that this is an unusual system: its polarized organization of junctions in the apical junctional complex is unlike that of other *Drosophila* epithelia and more like vertebrate epithelia; cells in this system polarize in a basal-to-apical direction; nor do these seem to involve canonical polarity factors.

Now they ask whether the cadherin adhesion system is necessary for polarization in the midgut, as it has been implicated in other epithelia. Their data indicate that the answer is no: the midgut polarizes largely normally when the E-cadherin system is disrupted and, although the cells also express N-cadherin, this is not compensating since polarization continues when both cadherins are targeted. The maneuvers do affect cell shape and are associated with expansion of the septate junctions, but these are relatively minor. The dominant result is that polarization doesn't seem to

require the classical cadherin system

Reviewer 1 Comments for the Author:

The analysis is logical and the data are high quality. These results are useful for the community. The weakness for me is that, in its current form, the manuscript is of rather specialist interest. It doesn't provide insight into how something as physiologically fundamental as epithelial polarity is being established without the input of "mainstream" mechanisms

The referee is correct that we are essentially reporting a negative result, i.e. that adherens junctions are not required for polarity in the midgut epithelium, but we think that this is important to publish, as this is the first example of an epithelium that does not require Cadherins to be correctly polarised. This means that it is lateral adhesion that is important for epithelial polarity and not some specific property of Cadherins.

It is not clear where such insight might come from, but possible directions include:

A) The study uses genetic approaches that will create sustained protein loss, so is it possible that more acute inhibition will reveal effects that are then compensated for (albeit by unknown mechanisms)?

We are unaware of any techniques for acutely depleting E-cadherin but would like to point out that we have depleted E-cadherin by RNAi using an enterocyte-specific driver that removes the protein after the cells have polarised, which is a more acute approach than generating mutant clones. Nevertheless, E-cad RNAi gave the same phenotype as *shotgun* mutant clones.

B) Could the key to the midgut system lie in the septate junctions? Is polarity affected if septate junctions are perturbed (or if septate junctions are perturbed in E-cadherin mutant animals)?

Indeed, in two previous papers from the lab (Chen et al. (2018) PLoS Biology; <https://doi.org/10.1371/journal.pbio.3000041>, Chen and St Johnston (2022) eLife, <https://doi.org/10.7554/eLife.76366>), we established that components of the integrin adhesion complex and septate junctions are required for the establishment of apical-basal polarity. Loss of the septate junction protein Mesh or the integrin adhesion component Talin results in almost 100% loss of epithelial polarity. Given the high penetrance of these phenotypes, we do not think that additionally depleting E-cadherin would be informative. Although adherens junctions are not required for the establishment and maintenance of apical-basal polarity in the midgut, their loss increases the length of the septate junctions and the levels of septate junction proteins (Fig. 4F). This suggests that loss of adherens junctions is largely rescued by strengthening the Septate junctions.

Reviewer 2 Advance Summary and Potential Significance to Field:

This study thoroughly explores the requirement for adherens junction components in the midgut

enterocytes of adult *Drosophila*. This epithelium is of particular interest for its unusual polarity mechanism relative to the rest of *Drosophila* epithelia, bearing similarity to the organization of junctions in vertebrates. The authors find that AJ mutant cells in the midgut, unlike in other studied epithelia, are still able to polarize and integrate seemingly normally. However, in the absence of a functional AJ, the SJ domain is found to expand. These findings are valuable to the polarity field, with implications for domain formation, and the broader cell biology community.

Reviewer 2 Comments for the Author:

My comments and suggestions are below.

- In Figure 1, the authors may wish to consider recoloring the color palette to (1) match the rest of the paper, and (2) be color-blind friendly.

We have now changed the colours in Figure 1 to match the rest of the paper.

- In Figure S3, the images of mutant clones appear to contain different numbers of cells across the different AJ mutants. Is there an effect on clone size apart from enterocyte shape? It would be of interest to include cell numbers per clone, or at least a comment in the text.

Clone size is quite variable in the gut but we don't see any obvious difference between the different genotypes, except for the *Ncad*^{D14} clones expressing RNAi as described earlier. We have now replaced the figure showing an a-catenin mutant clone with another that has a similar clone size to the other mutants.

- The use of different colors of data points in plots doesn't appear to be explained (e.g. Fig.4D-F and others). I assume the colors represent different guts, but this should be stated in the figure legend along with the number of guts for each sample.

Yes, these are SuperPlots, where the different colours represent data from different guts. We now state this in the legends and have included the citation to the paper describing Superplots in the Materials and Methods. We have also added the number of guts/animals used in each experiment in the figure legends and the total number of animals analyzed in the Methods.

- In Figure 4, the authors observe a significant increase in the length of SJs. Could the authors shed light on whether the increased length is due to existing proteins extending their domain, or an increased expression of SJ proteins?

Including a measurement of SJ intensity might be one approach.

We have now added a quantification of SJ staining intensity (*Mesh*, *Lgl*, *Scrib*, *Coracle* and *Tsp2A*) in the different mutants (Fig. 4F and Fig. S4F). The intensity of SJ staining per unit length is similar in mutant and wild-type enterocytes. However, since the Septate junctions are longer in *shg* and *arm* mutants, there is more total protein in the Septate Junctions in the mutants than in the controls. Thus, enterocytes either produce or localise more Septate junction proteins in response to the loss of adherens junctions to compensate for decreased cellular adhesion.

-The descriptive finding that polarity is maintained in gut enterocytes following E-cad RNAi was also reported in PMID 28847000 (Extended Data Fig 4); although the result was not quantitatively and thoroughly characterized as in this study, the authors may wish to consider citing it.

We apologize for this oversight. The paper was cited in the introduction but not next to the RNAi experiments, where we have now added it: "Downregulation of E-cadherin using two different RNAi lines led to a marked reduction in β -catenin localisation to cell junctions but did not cause any apical-basal polarity defects, as previously reported (Liang 2017)"

- In Figure 2A,C (and other figures), the use of the word 'BASAL' to describe the view from the bottom-up could be confusing (for example, SJ proteins are shown from this view), and the term is often used when apical-basal views are shown as in Figure 1. I might suggest leaving 'BASAL' out of the figures showing a bottom-up view, and rather simply describing it in the legend.

We now describe each of these panels as "A horizontal section through the basal region of a midgut containing a clone..."

- In Figure 6, the wording of "Ncadi" could be confusing. The authors may wish to use a more standard indication that RNAi is used, such as with superscript ("Ncad RNAi" or "Ncad KD", for example).

We have now replaced "Ncadi" with "*NCad1*^{RNAi}" in the graphics and text.

Reviewer 3 Advance Summary and Potential Significance to Field:

The manuscript by Carvalho et al. describes the drosophila midgut enterocytes as a unique system where, unlike the other fly epithelia, adherens junctions are localized below the septate junctions on the lateral cell membranes, which resembles the arrangement in the vertebrate epithelia where adherens junctions are important in organizing polarity complexes. The current work follows up on the previous findings from the lab (Chen et al. 2018), that midgut enterocytes lack the expression of key polarity molecules such as - Bazooka (Par-3), and they maintain their polarity independent of the canonical polarity system, thus asking how is E-cadherins still utilized in this tissue to establish polarity. Using genetics perturbations and spatiotemporal localization, they have shown that E-cadherins depletion in the epithelia results in basal extension of the septate junction boundary, reduction of the sub-septate junction lateral membrane domain, and allows the cells to become more circular, without affecting polarity. Further, they show that N-cadherin does not compensate for E-cadherin's loss of function. Altogether, this study indicates that the adherens junction regulates septate junction length but not apicobasal polarity in this endodermal origin epithelial system.

While the manuscript is written clearly, it was hard for me to understand what the key question is and what are the important takeaways from this study.

We state our key question at the end of the introduction:

"The distinct organisation of the midgut epithelium raises the question of what roles E-cadherin adhesion complexes play in enterocyte polarisation. E-cadherin has previously been shown to mediate adhesion between basal intestinal stem cells (ISCs) and their progeny enteroblasts (EBs), where it acts to delay EB detachment to allow Delta-Notch signalling to specify EB identity (Maeda et al., 2008). In addition, down-regulation of E-cadherin in apoptotic enterocytes triggers the release of epidermal growth factors to stimulate compensatory divisions of the ISCs, thereby maintaining homeostasis (Liang et al., 2017). E-cadherin must also be removed from the apical surface of integrating enteroblasts to establish an apical membrane initiation site (AMIS) (Chen and St Johnston, 2022). However, the effects of E-cadherin complex depletion on enterocyte polarity and differentiation have not been analysed. Here we set out to investigate the role of E-cadherin complex components in this process".

."

The important takeaways from this study are that:

1) *"the core components of the adherens junctions are not required for either the establishment or maintenance of apical-basal polarity in the midgut epithelium, unlike all other epithelia that have been examined so far".*

"Our results demonstrate that adherens junctions are not required for the organisation or

polarisation of the midgut epithelium. This contrasts with observations in other epithelia, where depletion of cadherin adhesion complexes disrupts epithelial organisation to produce mesenchymal-like cells and impairs apical-basal polarity (Bonello et al., 2021; Capaldo and Macara, 2006; Cox et al., 1996; Larue et al., 1994; Roy and Berx, 2008; Sheppard et al., 2022; Tunggal et al., 2005). This reinforces the evidence that the midgut epithelium polarises by a different mechanism from other Drosophila epithelia."

2) *" Although E-cadherin adhesion complexes are dispensable for polarity in the midgut, they do play a role in determining the relative sizes of the septate junction and sub-septate junction domains."*

In the current format, it is a series of interesting observations without providing insight into the underlying mechanisms and why these observations matter for tissue morphogenesis.

We would argue that it is important to show that adherens junctions play no role in the polarity of the midgut epithelium, since they are required for the polarity of all other well-characterized epithelia and are widely assumed to be a defining characteristic of epithelial tissues. While this is a negative result, it is nevertheless important as our results rule out the mechanisms that have been proposed to polarise other epithelia.

The results and discussion section can be further refined to discuss the current findings in the context of what's known and what it contributes towards further advancing the field. Further, the lab has already studied and established that the integrin adhesion complex-septate junction axis determines EC polarity (Chen et al. 2018). Now they find adherens junctions are dispensable for midgut enterocytes polarity.

We feel that we have discussed our results in the context of what is known and what it contributes towards further advancing the field. For example, as cited above, we discuss the fact that the midgut enterocytes are the only epithelial cells that have been shown to polarise in the absence of Cadherin adhesion complexes. We also discuss our finding that the septate junctions extend in Cadherin adhesion complex mutants as follows *"The expansion of the septate junctions in shg and arm mutants suggests that E-cadherin adhesion complexes somehow limit their basal extension. This could be by acting as a barrier that prevents septate junction proteins spreading more basally or by promoting their endocytosis at the boundary between the two domains. A third possibility is that septate junction proteins are up-regulated in shg and arm mutants, leading to lengthening of the septate junctions to compensate for the absence of Cadherin adhesion complexes. As is the case for the sub-septate junction domain, loss of α -catenin has no significant effect on the size of the septate junctions, making it unlikely that the linkage of E-cadherin to F-actin through α -catenin plays a role in establishing the border between the adherens and septate junctions"*.

However, how is this important for tissue development and morphogenesis is not clear.

We would like to point out that we are not studying the morphogenesis of the gut in a developmental context in this manuscript, since all our experiments were conducted in fully-developed adult midguts during homeostasis. However, one could consider cell height and the length of the septate and adherens junctions as morphogenesis at the cell scale.

Major comments -

1. It is not clear from the manuscript why limiting the septate junction is essential for this tissue. E-cadherins knockdown resulting in extended septate junctions, alters cell shape/aspect ratio but doesn't affect the polarity of the ECs and it also does not seem to perturb gross tissue morphology. What is the functional consequence of these changes in cell shape or septate junction length on tissue integrity and morphogenesis?

The changes in cell shape and septate junction length that we identified do not affect tissue integrity and morphology, but they may affect its function or resilience under stress, which are very difficult to assess. Downregulation of E-cadherin has also been shown to have no effect on intestinal permeability (PMID 28847000).

2. What mechanism is underlying E-cadherin control of septate junction size?

Authors have evoked the role of exocytosis /endocytosis, but can they functionally test this?

This would be an interesting question to investigate in future but is beyond the scope of the current work.

3. An interesting observation in the manuscript suggests- that loss of N-cadherin can stabilize β -catenin, possibly because of compensatory E-cadherin upregulation.

We have now removed this observation from the manuscript because we only observed this phenotype in some experiments and not in others. Since it is tangential to the main point of the manuscript and we have no explanation for why *Ncad1* mutants might affect Armadillo in the cytoplasm, we felt that these data detracted from the flow of the argument.

Another important finding in the manuscript is that N-cadherin can't compensate for E-cadherin. However, it is still not established whether N-cadherin has significant expression in this tissue, and it can be argued that the lack of expected response to N-cadherin knockdown is due to the lack of its expression. Thus, it would be important to show that N-cadherin is indeed expressed in the tissue and is downregulated upon RNAi treatment.

We have now examined the expression of endogenous N-cadherin 1::GFP in midguts. We observed very weak expression of *Ncad1* in wild-type ISCs, but this became much stronger when E-cadherin was depleted by *shg* RNAi. When *Ecad* was depleted more efficiently by raising the temperature to 29°C, we also observe *Ncad1*::GFP localisation to the lateral domain of the enterocytes, suggesting that *Ncad1* is up-regulated in response to the loss of *Ecad*. We have added these data as Figure 5A and mention the results in the text.

4. The authors find that upon loss of E- and N-cadherin, β -catenin accumulate in cytoplasm, eventually concluding that “N-cadherin plays a role in the destabilisation of β -catenin that is independent of E-cadherin”

How did they reach to this conclusion? What is the experimental evidence? And why this is relevant for the tissue?

As mentioned above, we have removed these data from the manuscript.

5. The first two figures are important descriptive results; however, I am finding it difficult to appreciate how they advance our knowledge. Also, these two figures can be easily combined into one figure.

We prefer to keep figures 1 and 2 separate for two reasons. 1) Since we go on to show that E-cadherin adhesion complexes do not perform the same function as they do in other epithelia, we think that it is important to confirm that they contain the same components as in other cells and form in the same way. 2) The two figures show different things. Figure 1 introduces the localisation of E-cadherin complexes below the septate junctions, which contrasts with all other *Drosophila* epithelia. Figure 2 shows that E-cadherin and β -catenin are mutually dependent on each other for their localisation, whereas α -catenin mutants have a weaker effect.

6. Authors have only mentioned the number of cell-cell junctions for all the quantitative analysis

plots. However, it will be useful to know the total number of animals used for quantification in each analysis.

The number of guts used for each analysis has been added to the figure legends and the total number of animals is now also included in the methods.

Second decision letter

MS ID#: jcs.263644R1

MS Title: Adherens junctions limit septate junction length in *Drosophila* midgut enterocytes but are not required for apical-basal polarity

Authors: Daniel St Johnston; Cátia A Carvalho; Mihoko A Tame
Article Type: BenchPress Revision

Dear Dr St Johnston,

I am happy to tell you that your manuscript has been accepted for publication in Journal of Cell Science, pending standard publication integrity checks.

Reviewer 1

SUMMARY OF THE ADVANCE MADE IN THIS PAPER AND ITS POTENTIAL SIGNIFICANCE TO THE FIELD

I think the authors have reasonably addressed the issues raised in my earlier review. I agree that some negative findings - such as the results of this report - are valuable for the community and should be published.

SUGGESTIONS TO AUTHORS

No further suggestions from me.

Reviewer 2

The authors have made a good effort to improve the manuscript, and it is now in a better state. I still think Figures 1 and 2 could be combined into a single figure. While they both present important descriptive results, I find it difficult to fully appreciate how these results advance our understanding. The Results and Discussion sections could be further refined to clarify the manuscript's contribution to the field and what new insights are being offered.

Reviewer 3

SUMMARY OF THE ADVANCE MADE IN THIS PAPER AND ITS POTENTIAL SIGNIFICANCE TO THE FIELD

The authors have satisfactorily addressed my previous comments, and the revised manuscript is an effective and detailed examination of the role of adherens junctions in the *Drosophila* midgut that will be of broad interest to the readers of JCS.

SUGGESTIONS TO AUTHORS

My sole, very minor suggestion: For figures showing clones, the authors indicate the genotype of the clone in a green font that may confuse some readers as it appears similar to the other figure

labels. The authors may wish to indicate the genotype for the figure in a clearer way, such as writing it above the figure and italicizing it.